

# Discovery of potential targets of Triptolide through inverse docking in ovarian cancer cells

Qinhang Wu, Gang Bao, Yang Pan, Xiaoqi Qian and Furong Gao

Department of Pharmacy, Nanjing University of Chinese Medicine, Nanjing, Jiangsu, China

## ABSTRACT

Triptolide (TPL) is proposed as an effective anticancer agent known for its anti-proliferation of a variety of cancer cells including ovarian cancer cells. Although some studies have been conducted, the mechanism by which TPL acts on ovarian cancer remains to be clearly described. Herein, systematic work based on bioinformatics was carried out to discover the potential targets of TPL in SKOV-3 cells. TPL induces the early apoptosis of SKOV-3 cells in a dose- and time-dependent manner with an $IC_{50} = 40 \pm 0.89$ nM when cells are incubated for 48 h. Moreover, 20 nM TPL significantly promotes early apoptosis at a rate of 40.73%. Using a self-designed inverse molecular docking protocol, we fish the top 19 probable targets of TPL from the target library, which was built on 2,250 proteins extracted from the Protein Data Bank. The 2D-DIGE assay reveals that the expression of eight genes is affected by TPL. The results of western blotting and qRT-PCR assay suggest that 40 nM of TPL up-regulates the level of Annexin A5 ($6.34 \pm 0.07$ fold) and ATP syn thase ($4.08 \pm 0.08$ fold) and down-regulates the level of β-Tubulin ($0.11 \pm 0.12$ fold) and HSP90 ($0.21 \pm 0.09$ fold). More details of TPL affecting on Annexin A5 signaling pathway will be discovered in the future. Our results define some potential targets of TPL, with the hope that this agent could be used as therapy for the preclinical treatment of ovarian cancer.

## INTRODUCTION

Ovarian cancer is currently a leading cause of mortality among gynecological malignant tumors (*Crane & Brown, 2018*). Ovarian cancer is derived from the ovarian surface epithelial layer surrounding the ovary and has a high rate of metastasis (*Vetter & Hays, 2018*). The introduction of combination therapy using paclitaxel and cisplatinum-based drugs can dramatically improve survival rates (*Wilson et al., 2018*; *Moore et al., 2018*). However, traditional chemotherapy drugs have significant adverse side effects and cause to drug resistance of most patients who are initially responsive to this therapy (*Moore et al., 2018*). Therefore, it needs to discover innovative approaches for the treatment of endometrial and ovarian cancer.

Numerous natural products extracted from plants have been proved to be potent compounds to suppress cancer cells (*Cevatemre et al., 2018*); for example, the clinical

Corresponding author
Qinhang Wu, wuqinhang@163.com

anticancer drugs of paclitaxel (*Nakamura et al., 2019*), etoposide (*Breitbach et al., 2019*), camptothecin (*Amin et al., 2018*) and vincristine (*Hafazalla et al., 2018*). Triptolide (TPL), a diterpene trioxide, is a key component of traditional Chinese medicine *Tripterygium wilfordii Hook. F.* (also known as Thunder God Vine) (*Chen, 2001*). It is biologically active against autoimmune diseases, inflammation and various human malignant tumors by inhibiting the proliferation of cancer cells (*Zhang et al., 2017*). TPL is also proved to be more effective than cisplatinum, taxol and camptothecin in inhibiting xenograft growth of some solid tumors. Researchers have reported to us that TPL could be considered as a potential chemotherapeutic for ovarian cancer (*Westfall, Nilsson & Skinner, 2008*; *Li et al., 2010*). However, its antitumor activity against gynecologic carcinomas has not yet been well investigated.

One hypothesis that TPL covalently modifies proteins via epoxide ring-opening reactions has been proved that how TPL stimulates apoptosis of cancer cells (*Hu et al., 2016*). Even though the target proteins are undefined, it is evident that TPL has some kind of explained effects on the anticancer. However, with the discovery of other mechanisms, TPL can suppress kinases or suggest epigenetic modifications through alkylation (*Sun et al., 2017*; *Zhong et al., 2013*; *Song et al., 2017*; *Liu et al., 2017*). *Chang et al. (2001)* have reported that TPL blocked p53-mediated cell cycle arrest to enhance doxorubicin-mediated apoptosis of tumor cells. Other studies have proved that NFκB, a potential target of TPL, can sensitize TNF-related apoptosis-inducing ligand-induced apoptosis in lung cancer cells (*Lee et al., 2002*). The connection between the antitumor activity of TPL and ovarian cancer remains to be discovered with novel methods.

We sought to use a new bioinformatics method to identify potential targets of TPL in SKOV-3 cells (one of the ovarian cancer cell lines) combining the bioassay and inverse docking (*Kamper et al., 2006*; *Furlan, Konc & Bren, 2018*) to save energy and money. Herein, inverse docking is carried out in the first step to screen proteins extracted from protein/ligand complexes, which are downloaded from Protein Data Bank (PDB, https://www.rcsb.org) (*Berman et al., 2000*; *Burley et al., 2019*; *Westbrook et al., 2003*). On the one hand, the binding sites can be generated automatically using the docking algorithm by identifying the place where native ligands place; on the other hand, it can be defined by reviewing the kinds of literature involved.

Firstly, SKOV-3 cells are incubated for 48 h to test the anti-proliferative activity of TPL. Secondly, the inverse docking protocol is employed to screen anti-cancer targets of TPL. Thirdly, a 2D-DIGE assay is utilized to further identify targets by testing differential gene expression in TPL-treated SKOV-3 cells from those screened proteins by inverse docking. Finally, western blotting and the qRT-PCR assay define the potential targets at the molecular level.

# MATERIALS AND METHODS

## Drug preparation

TPL (purity 99.0%, Beijing beihua hengxin biotechnology Co., Ltd., Beijing, China) and cisplatinum (P4394, Sigma-Aldrich Co. LLC, St. Louis, MO, USA) were dissolved in

dimethyl sulfoxide (DMSO) to a stock concentration of 500 nM. The solution was stored at −20 °C and diluted in a cell culture medium.

## Cell culture

Human ovarian cancer cell line SKOV-3 (ATCC: HTB-77) were cultured on cell plates at 37 °C, in 5% $CO_2$, in RPMI-1640 (HyClone) supplemented with 10% fetal bovine serum, 100 units/ml penicillin and 0.1 mg/ml streptomycin.

## MTT assay

SKOV-3 cells were plated in the 96-well plates ($5 \times 10^3$/well). RPMI-1640 culture medium was used as the blank control group. A final concentration of 0, 3, 6, 12, 24, 48, 96,192 and 384 nM of TPL and 0, 1.25, 2.5, 5, 10, 20, 40, 80 and 160 μM of cisplatinum (used as the positive control) were added to the wells for 24, 48 and 72 h. Five duplicates were set for each concentration. MTT (20 μl, 5 mg/ml, Sigma, St. Louis, MO, USA) was added to each well at 37 °C for 4 h. Then 150 μL of DMSO was added to each well after the liquid supernatant was removed. The enzyme-linked immunosorbent assay reader (TECAN, SPARK 10M; Mannedorf, Switzerland) was used to obtain the results at 570 nm. The $IC_{50}$ values of TPL and cisplatinum were calculated by SPSS Software (IBM Inc., New York, USA, version 21.0).

## Apoptosis assay

After harvested and washed twice SKOV-3 cells were stained with 2 μM JC-1 for 15 min at 37 °C. A flow cytometer using 480 nm excitation and 585 nm emission filters was applied to analyze the stained wells. Cell suspension mixed with 10 μM Carbonyl cyanide m-chlorophenyl hydrazine (CCCP) was set as the control. The cell mixture was incubated for 15 min at 25 °C in the dark followed by fluorescence-activated cell sorting (FACS) cater-plus flow cytometry (FACSAria III, BD CellQuest pro).

## Inverse docking

Classical biological methods to find the antitumor targets of natural products usually consume manpower and material resources. New approaches such as inverse docking are developed to efficiently fish the potential targets of natural products like resveratrol (*Kores et al., 2019*), cassiar in alkaloids (*Negi et al., 2018*) and curcumin (*Furlan, Konc & Bren, 2018*). To identify the potential targets of TPL, we developed an inverse docking protocol (*Bren, Fuchs & Oostenbrink, 2014*) by using a GOLD program (Genetic Optimization for Ligand Docking, version 3.5) (*Jones et al., 1997*) in combination with Discovery Studio (DS, BIOVIA[TM], Rueil-Malmaison, France, version 4.0).

TPL was subjected to DS as the template ligand. The target library was built based on 2,250 protein/ligand complexes extracted from PDB (supplied by BIOVIA[TM], updated to 2015). Each protein was standardized with DS Prepare Protein protocol where missed loops were inserted, protonation was performed for pH of 7.4 with the "Protein Ionization" method, and CHARMm forcefield was used for calculations while native ligands, as well as water, were deleted. Then, the pockets where the native ligands placed were defined as the binding sites. The radius of the docking-grid sphere was set to 1.5 nm to

accommodate the entire TPL molecule as long as at least one of its atoms is placed less than 0.6 nm away from the binding sites. The inverse docking protocol applied semi-flexible docking based on a genetic algorithm to save the CPU time and acquire accurate docking results (*Xue et al., 2013*). The algorithm sets the ligand and binding sites flexible to generate 255 poses while the other region of the protein rigid. The inverse docking search protocol could successfully dock TPL into a candidate protein before all possible positions and orientations of 255 poses were exhausted. At most five poses were stored from every docking simulation, except if the best three docking poses had root-mean-square deviations (RMSD) smaller than 0.15 nm. The X ray_Score calculated by GoldScore and Ludi_score was used to score and rank the poses of inverse docking (see in Supplemental File).

## 2D gel electrophoresis

SKOV-3 cells treated with TPL were washed twice with ice-cold PBS and resuspended in lysis buffer for 30 min to collect the cell suspension. After incubated for 15 min at 25 °C, samples were mixed with *N. N*-dimethylacrylamide for alkylation and DTT for centrifugation. Protein samples were separated by 2D gel electrophoresis using the first dimension, isoelectric focusing (IEF) and the second dimension, non-equilibrium pH gradient electrophoresis (NEPHGE). The first dimension was performed with a nonlinear pH 3–10 range at 25 °C in IEF buffer (0.5% Pharmalyte 3–10 NL, 2% (w/v) ASB, 15 mM DTT, 2 M thiourea, 6 M urea and bromophenol blue) containing 0.16 mg labeled proteins. After IEF, the immobiline strip with separated proteins was equilibrated for 15 min using equilibration buffer (Thermo Fisher Scientific, Waltham, MA, USA). Separated protein samples were then further separated on 4–12% polyacrylamide gel. After labeling with Cy3- or Cy5- fluorescent dye, gels were scanned on a 2920 2D-Master Imager (Bio-Rad, Hercules, CA, USA) according to manufacturer's instruction. Protein spots were migrated and detected automatically, then quantified by calculating the fluorescence intensities using DeCyder Differential In-Gel Analysis Software (Bio-Rad, Hercules, CA, USA).

## Peptide mass fingerprinting

Protein spots excised from the gel were destained with trypsin (Sigma Aldrich, St. Louis, MO, USA) then incubated in MilliQ-H$_2$O for 10 min to cut into small pieces. Peptides were extracted twice from supernatants of the disposed of gel pieces with 50% (v/v) acetonitrile and 0.1% (v/v) trifluoroacetic acid. After pooled and dried, extracts were analyzed using a TofSpec-2E mass spectrometer (Micromass, Manchester, UK). Peptide mass fingerprinting maps were acquired and screened in the Swiss-Prot database performed by the Mascot Software.

## Western blotting

Cells were washed twice and lysed in ice-cold RIPA lysis buffer for 30 min to collect the cell suspension. After separated with SDS-PAGE, cell lysate aliquots (40 μg) were blotted onto a polyvinylidene difluoride membrane and incubated with primary antibody, rabbit

polyclonal anti-Annexin A5 (diluted 1:300; Abcam), anti-ATP synthase (diluted 1:300; Abcam), anti-β-tubulin (diluted 1:300; Abcam) and anti-HSP90 (diluted 1:300; Abcam) overnight at 4 °C. Membranes were washed and incubated with goat anti-rabbit IgG horseradish-peroxidase (HRP) pre-adsorption secondary antibody (1:2,000; ab7090, Abcam, Cambridge, UK). Bands were visualized by an enhanced chemiluminescence system (ECL, Abcam, Cambridge, UK), then quantified using the manufacturer protocol (Bio-Rad, Hercules, CA, USA). The PowerPacTM HC electrophoresis instrument and film transfer instrument (Bio-Rad, Hercules, CA, USA) was used in this section.

### RNA isolation and quantitative real-time PCR

Total RNA was extracted from SKOV-3 cells treated with TPL (0, 10, 20, 40 nM for 48 h) using the RNA extraction kit (Invitrogen, Life Technologies, Carlsbad, CA, USA). cDNA was synthesized from 2 μg of total RNA using Superscript reverse transcriptase (Life Technologies, Carlsbad, CA, USA). The primers used for quantitative real-time PCR (qRT-PCR) were as follows: for HSP90, sense primer (5′-TTAAGGTACTACACATCTGCCTC T-3′) and antisense primer (5′-TGCTTTCGGAGACGTTCCACAA-3′); for Annexin A5, sense primer (5′-CAGTCTAGGTGCAGCTGCCG-3′) and antisense primer (5′-GG TGAAGCAGGACCAGACTGT-3′); for ATP synthase, sense primer (5′-TCTTTGCTGG TGTTGGTGAA-3′) and antisense primer (5′-TGAGCTCATCCATACCCAAA-3′); for β-tubulin, sense primer (5′-TGCATTGACAACGAGGC-3′) and antisense primer (5′-CTGTCTTGACATTGTTG-3′). The housekeeping gene glyceraldehyde phosphate dehydrogenase (GAPDH) was used for normalization (sense: 5′-TGATGACATCAAG AAGGTGGTGAAG-3′; and antisense: 5′-TCCTTGGAGGCCATGTG GGCCAT-3′). PCR products were analyzed by 1.2% agarose gel electrophoresis with ethidium bromide for UV light transilluminator visualization using LightCycler® 96 qRT-PCR instrument (Roche Life Science, Penzberg, Germany).

### Statistical analyses

Statistical analysis was performed by using one-way analysis of variance, followed by Dunnett's Multiple Comparison $T$-test using SPSS Software (IBM Inc., New York, USA, version 21.0) when appropriate. All data were presented as mean ± standard deviation (SD). The difference was considered statistically significant at $^*p < 0.05$ and $^{**}p < 0.01$.

## RESULTS

### Evaluation of the anti-cancer effect of TPL on SKOV-3 cell line

The efficacy of TPL on ovarian cancer and its target genes have not been deeply studied. To test the ability of TPL to inhibit the survival of ovarian cancer cell in vitro, cell viability assay using the SKOV-3 cell line was performed. Compared to cisplatinum, TPL exhibits a better potency with $IC_{50}$ values of the nanomolar level. Significant suppression of proliferation in SKOV-3 cells is observed after treatment with TPL at the concentrations of 3, 6, 12, 24, 48, 96,192 and 384 nM respectively (Fig. 1). When the TPL dose increased, a gradual decrease in cell viability is observed either cells are treated

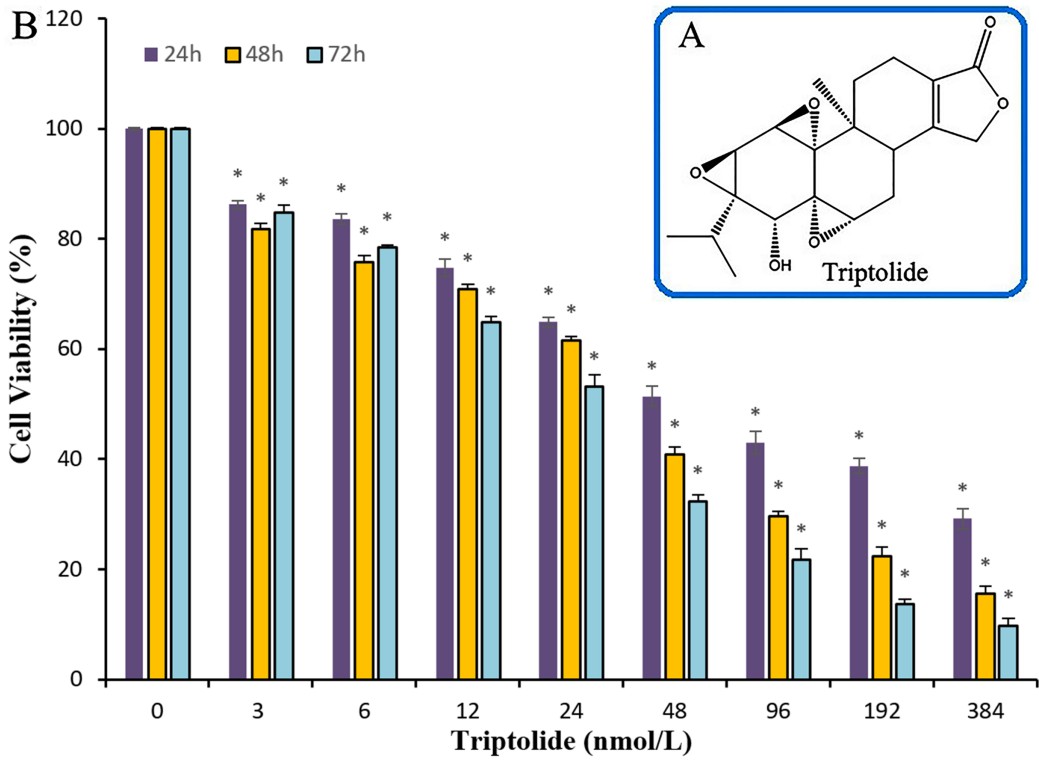

**Figure 1 The chemical structure of TPL and the histogram for the anti-cancer effects of TPL on SKOV-3 cell in line.** (A) The chemical structure of TPL. (B) Effects of TPL on the growth of endometrial and SKOV-3 cell in vitro. *$p < 0.05$ vs. control.

**Table 1 The IC$_{50}$ value of different drugs on SKOV-3 cells of TPL.** The IC50 value of cisplatinum and triptolide.

| Time (h) | IC$_{50}$[a,b] | |
| --- | --- | --- |
| | Triptolide (nmol/L) | Cisplatinum (μmol/L) |
| 24 | 70.3 ± 1.17 | 33.0 ± 1.77 |
| 48 | 40.0 ± 0.89 | 19.2 ± 1.32 |
| 72 | 31.7 ± 1.23 | 9.6 ± 0.74 |

Notes:
[a] Values are means of three experiments.
[b] IC$_{50}$, compound concentration required to inhibit tumor cell proliferation by 50%.

for 24, 48 or 72 h. At the concentration of 3 and 6 nM, little significant time dependance of cell treatment for 24, 48 or 72 h was observed. Along with the increasing concentration, the time dependance becomes clear among 24, 48 and 72 h. It can also be seen in Table 1, TPL exhibits the IC$_{50}$ value of 70.3 ± 1.17 nM and 31.7 ± 1.23 nM on 24 and 72 h. Meanwhile, cisplatinum exhibits larger IC$_{50}$ value of 33.0 ± 1.77 μM and 9.6 ± 0.74 μM on 24 and 72 h. All the data show that TPL inhibits ovarian cancer cell proliferation more potently than cisplatinum in a dose-dependent manner in SKOV-3 cell lines.

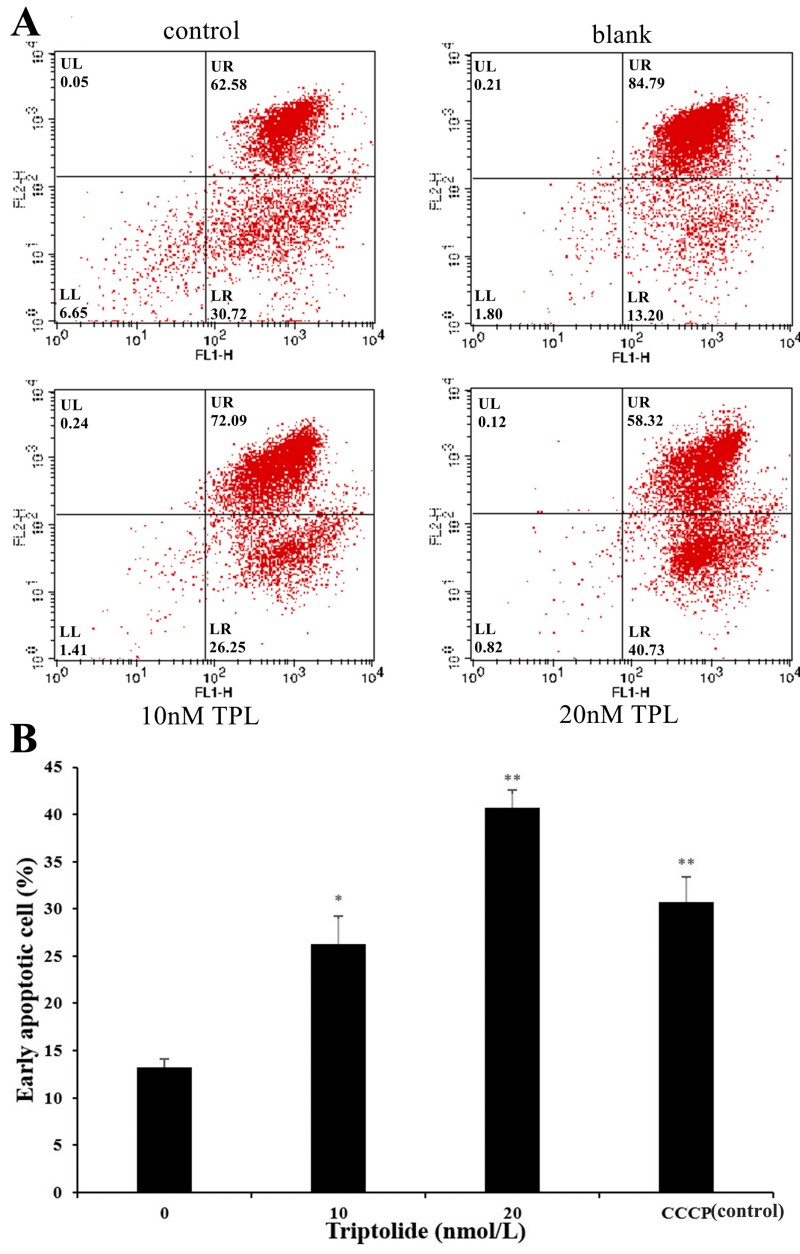

**Figure 2 Flow cytometric analysis of SKOV-3 cells using the MitoProbe JC-1 Assay Kit.** (A) Flow cytometric analysis of SKOV-3 cells using the MitoProbe JC-1 Assay Kit. (B) The histogram of the results of flow cytometric analysis. DMSO as the blank, CCCP (10 μM) as the control, TPL (10 nM) and TPL (20 nM) on the apoptosis of SKOV-3 cells. *$p < 0.05$ vs. control, ** $p < 0.01$ vs. control.

## Apoptotic changes in TPL-treated ovarian cancer cell

As the increased doses of TPL, we also detected a simultaneous increase in both LR fraction (early apoptosis) and LL fraction (regarded as necrotic) subpopulations. As is shown in Fig. 2A, the degree of apoptosis is quantitatively expressed as the control (DMSO) presented 13.20% population after 24 h treatment. The apoptosis rate of SKOV-3 cell rises after the treatment of TPL. Both the low and high concentrations of TPL significantly

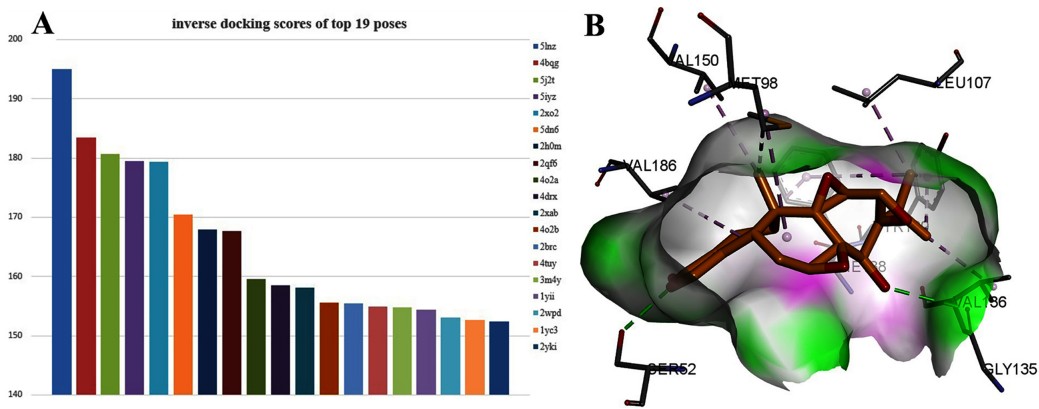

**Figure 3** **The 19 proteins screened by inverse docking. The docking pose of TPL binding to HSP90.** (A) Inverse_docking scores calculated by GOLD of top 19 proteins. (B) The docking pose of TPL binding to HSP90 (PDB ID: 5LNZ). The binding pocket of HSP90 protein displayed as a α-helix with a surface. All compounds are shown with only backbone atoms. 

promote the apoptosis of SKOV-3 cells with the early apoptosis rate of 26.25% and 40.73% (Fig. 2B). In contrast to low concentration, a high concentration of TPL slightly enhances the apoptosis of SKOV-3 cells and is higher than those treated with 10 µM CCCP. A dose-dependent increase of early apoptosis can also be seen from the results after TPL treatment of SKOV-3 cells. The results of the apoptosis assay indicate that TPL inhibits SKOV-3 cells by inhibiting certain targets rather than by killing cytotoxic SKOV-3 cells.

## Inverse docking

Interestingly, TPL is more active than cisplatinum. To find potential target genes of TPL, an inverse docking protocol *in silico* was designed based on GOLD. With ranked X ray_Score values calculated by the GOLD scoring protocol, the top 19 poses with the X ray_Score values larger than 150 were selected (see in Supplemental Files). Among them, seven proteins (5lnz, 4bqg, 2qf6, 2xab, 2brc, 2xab, 2yki) belong to HSP90, six proteins (5j2t, 5iyz, 4o2a, 4drx, 4o2b, 4tuy) belonged to β-Tubulin, three proteins (2xo2, 2h0m, 1yii) belong to Annexin A5 and three proteins (5dn6, 3m4y, 2wpd) belong to ATP synthase respectively (Fig. 3A). HSP90 protein has the highest docking scores compared with the other three proteins. As in Fig. 4B, two hydrogen bonds between TPL and HSP90, Ser52 and Gly135 ensure the correct binding pose of TPL, as well as hydrophobic interactions, stabilize the binding mode. However, the amount of HSP90 proteins are not significantly different from other proteins. Therefore, further biological experiments are necessary to identify the potential targets for TPL.

## Differential gene expressions in TPL-treated SKOV-3 cell

To further determine the main targets of TPL on ovarian cancer cells, a 2D-DIGE assay was utilized to examine global changes in gene expression in SKOV-3 cells after treatment with 10nM TPL for 48 h. Total proteins from SKOV-3 cell were minimally labeled with two different fluorescent dyes, mixed and separated by isoelectric focusing on a 2D gel on SDS-PAGE. The identical proteins migrating to the same 2D spots were detected and

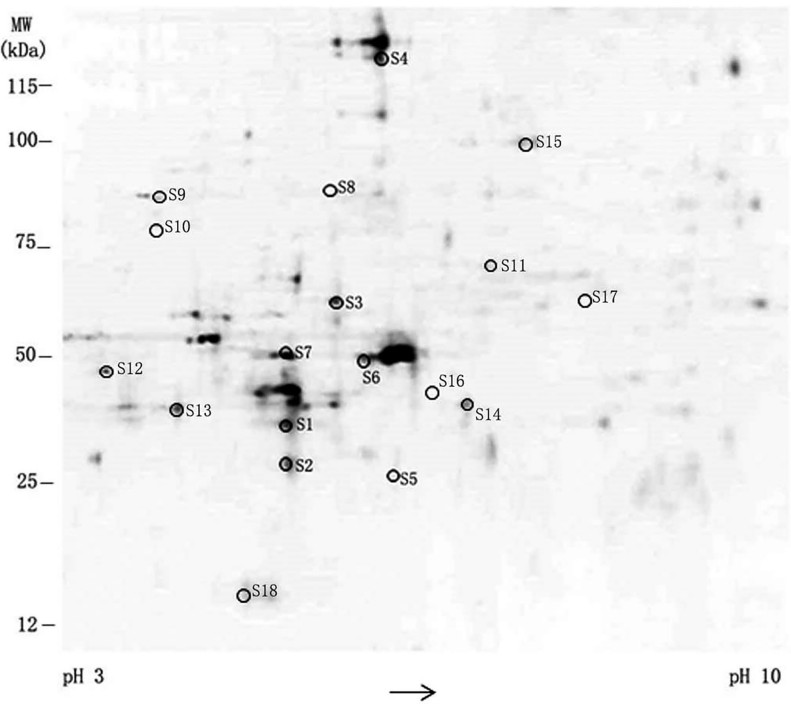

**Figure 4 The results of differential gene expression in TPL-treated SKOV-3 cell.** Differential expression profile of SKOV-3 proteome between 0 and 10 nmol/L TPL based on 2D-DIGE.

quantified based on their fluorescence intensities to compare their expression by the DeCyder program.

The difference of spots' density between normal and TPL-treated SKOV-3 cells discloses the changes of expression in eighteen protein spots (Fig. 4). Among 18 protein spots, the density between TPL and control is larger than two in eight spots. They are chosen to acquire finger printers of the peptide mass via MS/MS spectrum. According to the database searching results, the eight proteins are verified as Annexin A5, Tropomyosin (alpha-3 chain), ATP synthase (subunit beta), POTE ankyrin (domain family member E), Rho (GDP-dissociation inhibitor), Actin (cytoplasmic), Tubulin (β-4B chain) and Heat shock protein 90-alpha (HSP90). The representative genes found to be up- and down-regulated by 10 nM of TPL treatment are summarized in Table 2, respectively. Their known or proposed functions are analyzed by Gene Ontology. These eight proteins are mostly involved in maintaining the cellular structure, signal transduction and transport.

## Effects of TPL on the expression of novel proteins

To further elucidate the mechanism of TPL in ovarian cancer and define the main target gene, western blotting (WB) and qRT-PCR analysis were performed on the four proteins to detect their differential expression. The results (shown in Fig. 5 and Table 3) reveal that TPL markedly up-regulates the levels of Annexin A5 and ATP synthase proteins as well as down-regulates the levels of β-Tubulin and HSP90 in ovarian cancer cell lines.

**Table 2  Up-regulated and down-regulated genes after treatment with TPL in SKOV-3 cells.** Differential gene expression in TPL-treated SKOV-3 cell. There are the known or proposed function of the identified proteins.

| Spot no. | Identified protien | Aession no. | Mr/pI | Ratio (TPL/ Control) | Known or proposed function |
|---|---|---|---|---|---|
| Up-regulated proteins | | | | | |
| S1 | Annexin A5 | gi\|4502107 | 35936.8/4.93 | 3.28 | anticoagulation and apoptosis |
| S2 | Tropomyosin alpha-3 chain | gi\|669633291 | 28720/4.69 | 2.27 | regulate the stability of actin |
| S3 | ATP synthase subunit beta | gi\|32189394 | 56559.9/5.26 | 2.55 | a key enzyme in bioenergetics of a living cell |
| S4 | POTE ankyrin domain family member E | gi\|134133226 | 121363.4/5.83 | 2.04 | regulate signaling between the inside and outside of cells |
| S5 | Rho GDP-dissociation inhibitor | gi\|961818162 | 22974/5.56 | 2.13 | inhibit cell carcinogenesis, infiltration and metastasis |
| S6 | Actin, cytoplasmic | gi\|45382927 | 41736/5.29 | 2.06 | maintain cell morphology |
| Down-regulated proteins | | | | | |
| S7 | Tubulin beta-4B chain | gi\|5174735 | 49831/4.79 | −2.39 | cell division |
| S8 | Heat shock protein HSP 90-alpha | gi\|154146191 | 84659.7/4.94 | −3.05 | maintain protein conformation |

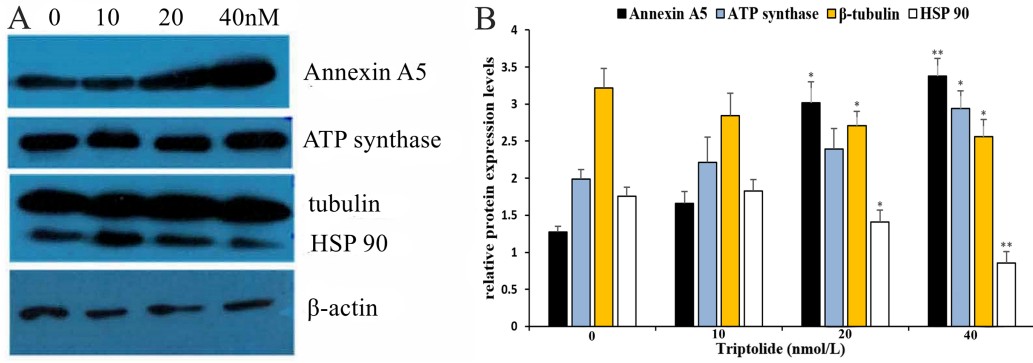

**Figure 5  Validation of differential expression proteins by western blotting of TPL.** (A) Validation of differential expression proteins by western blotting. (B) The histogram of the relative protein expression levels (equates to protein/β-actin). $^*p < 0.05$ vs. control, $^{**}p < 0.01$ vs. control.

WB results show that the expression of Annexin A5 and ATP synthase increases in a dose-dependent manner after TPL treatment for 24 h. According to qRT-PCR assay results, TPL stimulates the expression of Annexin A5 at 40 nM, especially more dramatically than ATP synthase ($6.34 \pm 0.07$ fold vs. $4.08 \pm 0.08$ fold). On the other hand, TPL inhibited β-Tubulin and HSP90 significantly more than 50% at the concentration of 20 and 40 nM. When cells are treated with 40 nM TPL. The relative expression of β-Tubulin mRNA decreases from 1 to $0.11 \pm 0.12$ fold.

# DISCUSSION

TPL, a pure compound extract from the Chinese traditional medicine *T. wilfordii*, has significant cytotoxic effects on ovarian cancer (*Hu et al., 2016*). However, the mechanisms of TPL on ovarian cancer and its target genes have not been studied (*Westfall, Nilsson & Skinner, 2008*). Herein, systematic work was performed to discover some transcription

**Table 3 Expression amount of relative mRNA in SKOV-3 cells treated with TPL at qRT-PCR assay.**
The results reveals that TPL markedly up-regulates the levels of Annexin A5 and ATP synthase proteins as well as down-regulates the levels of β-Tubulin and HSP90 in ovarian cancer cell lines.

| Triptolide (nmol/L) | Relative expression amount of mRNA | | | |
|---|---|---|---|---|
| | Annexin A5 | ATP synthase | β-Tubulin | HSP 90 |
| 0 | 1 | 1 | 1 | 1 |
| 10 | 1.55 ± 0.06** | 1.37 ± 0.15* | 0.76 ± 0.13 | 0.67 ± 0.08 |
| 20 | 3.68 ± 0.04** | 2.79 ± 0.11** | 0.38 ± 0.19* | 0.42 ± 0.06** |
| 40 | 6.34 ± 0.07** | 4.08 ± 0.08** | 0.11 ± 0.12** | 0.21 ± 0.09** |

Notes:
* $p < 0.05$ vs. control.
** $p < 0.01$ vs. control.

factors in SKOV-3 cell when treated with TPL. In contrast to cisplatinum, TPL significantly increases the number of apoptotic cells in ovarian cancer cell lines. More than 50% of cell apoptosis is observed when the SKOV-3 cell is treated with 48 nM for 48 h. Besides, the annexin V assay can detect that 40 nM TPL dramatically promotes early apoptotic cells at the rate of 40.73%. It means that apoptosis slowly induces in SKOV-3 cells. These results suggest that TPL could be investigated as a possible therapeutic agent for ovarian cancer. However, the mechanism by which TPL exerted its anticancer activities and the target genes TPL up- or down-regulated remains unclear.

To find its potential target from many proteins, an inverse docking protocol was developed based on the GOLD program, using TPL as the template ligand to fish the well-docked proteins downloaded from PDB. Four targets from the top 19 proteins are selected with the docking scores larger than 100. However, there is no enough difference between them. To find out more details of TPL acting on the four target genes, the gene expression profile changes in SKOV-3 cells are investigated after the treatment with TPL. Four proteins with the spots' density larger than 2.5 in the 2D-DIGE assay are chosen to quantitatively determine changes in their relative mRNA expression induced by TPL treatment. In SKOV-3 cell treated with 40 nM TPL, mRNA levels of Annexin A5 and ATP synthase are up-regulated to 6.34 ± 0.07 and 4.08 ± 0.08 fold, while mRNA levels of β-Tubulin and HSP90 are down-regulated for 0.11 ± 0.12 and 0.21 ± 0.09 fold.

Initially, TPL has been demonstrated to cause global transcriptional inhibition via the largest subunit of RNA polymerase II (*Zhao et al., 2012*). Recently, TPL is believed to target specific transcription factors, such as Bcl-2, Bcl-xL, NF-kB, pRB, RNA polymerases (*Wu et al., 2014*; *Zhang, Ho & Wong, 2018*; *Li et al., 2017*; *Guan et al., 2017*; *Kim & Park, 2017*). In this study, we provide evidence that TPL inhibits β-tubulin and HSP90. Several natural products including paclitaxel, colchicine and vinca alkaloids have been proved to bind β-tubulin at a distinct binding site (*Zhou & Giannakakou, 2005*). As a natural product isolated from the medicinal plant, TPL effectively reduces β-tubulin, which means that TPL can inhibit ovarian cancer cells via down-regulation of β-tubulin. However, more details of the pathway involving β-tubulin remain unclear and the binding sites on β-tubulin bound by TPL have yet to be confirmed. As a cellular protein in the annexin group, Annexin A5 binds to anionic phospholipids with high affinity, although

the detail function of the protein is unclear. However, Annexin A5 has been proved to be an essential protein in the inhibition of blood coagulation. Recent studies have shown that Annexin A5 inhibits the activity of phospholipase A1 by competing with prothrombin for phosphatidylserine binding sites (*Zhou et al., 2018*). All of these details revealed in several in vitro studies suggest that Annexin A5 plays a key role in the subsistence and apoptosis of cancer cell (*Becarevic et al., 2018*; *Su et al., 2018*; *Sun et al., 2018*). The further experiment suggests that Annexin A5 significantly regulates the signaling pathway involved in cell apoptosis on ovarian cancer (*Becarevic et al., 2018*). This evidence inspires us that the potential target of TPL effect on ovarian cancer cells could be Annexin A5. More details on how TPL affects Annexin A5 signaling will be revealed in the future.

## CONCLUSION

We first used the inverse docking *in silico* to discover targets of TPL at cellular and molecular levels. Although there exist more details about how TPL inhibits SKOV-3 cells, our research suggests that it could be investigated as a possible therapeutic agent for ovarian cancer. Four target genes, Annexin A5, ATP synthase, β-tubulin and HSP90, are presented here to be regulated significantly by TPL treatment in SKOV-3 cells. Further work will focus on the signaling pathways which are involved in ovarian cancer.

### Funding

This research was supported by the Doctoral Fund of Ministry of Education of China (20133237120011) and a project funded by the Priority Academic Program Development of Jiangsu Higher Education Institutions (PAPD). The funders had no role in study design, data collection and analysis, decision to publish, or preparation of the manuscript.

### Grant Disclosures

The following grant information was disclosed by the authors:
Doctoral Fund of Ministry of Education of China: 20133237120011.
Priority Academic Program Development of Jiangsu Higher Education Institutions (PAPD).

### Competing Interests

The authors declare that they have no competing interests.

### Author Contributions

- Qinhang Wu conceived and designed the experiments, analyzed the data, authored or reviewed drafts of the paper, and approved the final draft.
- Gang Bao performed the experiments, prepared figures and/or tables, and approved the final draft.
- Yang Pan conceived and designed the experiments, analyzed the data, authored or reviewed drafts of the paper, and approved the final draft.

- Xiaoqi Qian performed the experiments, prepared figures and/or tables, and approved the final draft.
- Furong Gao performed the experiments, prepared figures and/or tables, and approved the final draft.

## Data Availability

The raw measurements are available in the Supplemental File.

## Supplemental Information

Supplemental information for this article can be found online at http://dx.doi.org/10.7717/peerj.8620#supplemental-information.

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
