# Peer review of "Discovery of potential targets of Triptolide through inverse docking in ovarian cancer cells"

_PeerJ, doi:10.7717/peerj.8620_

## Round 0.1 · original submission · Major Revisions

All three reviewers recommended a major revision in the manuscript. Particular attention should be devoted to English usage. The authors carried out statistical analysis using one-way analysis of variance, followed by Dunnett's Multiple Comparison t-test using SPSS Software (IBM Inc., New York, USA). In my view, the methodology is adequate. The authors should also address the following point.

In the materials and methods, the authors say, “Targets library were built on proteins extracted from PDB (Protein Data Bank, https://www.rcsb.org) and the binding sites of those proteins were defined where their native ligands placed.” Please provide at least one reference for the protein data bank. For instance, J. Westbrook, Z. Feng, L. Chen, H. Yang, H. M. Berman, Nucleic. Acids. Res. 2003, 31, 489–491.

Reviewer 1 ·

Basic reporting

1) The English used is very poor, which often hinders the understanding of the manuscript. The manuscript should be reread and corrected by a person skillful in scientific English writing. Past perfect should be overall replaced by present perfect.

2) Reference to manuscript J. Chem. Inf. Model. 2019, 59, 2467−2478 should be given in the inverse docking section.

3) Figure 1 is of extremely low quality. It should be replaced by one having a better resolution.

Experimental design

The molecular docking protocol is described in an irreprodicible fashion. Use Chem. Res. Toxicol. 2014, 27, 2136−2147 as a template and quote it.

Validity of the findings

No comment.

Additional comments

Chemical structure of triptolide should be provided in a separate figure.

Reviewer 2 ·

Basic reporting

-- The manuscript is full of typos and syntax(even in the affiliation of the authors which should be Pharmacy not Parmacy) , It should be checked by an English native speaker to improve the quality of the used language.
-- Good background and suitable references
-- Article structure should be revised, It will be better to start with the reverse docking and then based on the results from the reverse docking , it could flow to the in vitro testing but not the reverse
-- The results are relevant

Experimental design

-- Original research within the scope of the journal.
The research question is well defined and relevant but the flow of the work is not logic concerning the in silico study, it should be the first step before paying effort and money and it should be used as guide for the subsequent work.
-- Methods are described in details except in the docking study more unwanted facts and details about the principle rather than the method itself were given. more details and references should be given for all the methods used

Validity of the findings

-- Most of the biological results have no statistical analysis and the level of significance and differences from the control are missed.

Additional comments

The manuscript: Discovery of potential targets of triptolide through inverse docking in ovarian cancer cells is interesting however there are many points that required more attention by the authors
among these points:
1- Since the abstract might present the only source of knowledge for many researchers, more rigid and real results should be given instead of vague sentences as it cause upregulation in ..... a measurable value should be given
2- From the title, it seems that inverse docking was used to discover potential targets in ovarian cancer cells then it was used to direct the in vitro study but the design was reversed, it seems that the authors start the in vitro testing then apply the reverse docking (the link between the in silico and in vitro testing was totally missed)
3- The purity of the triptolide, the specs of all the devices used and the version of the used software are not found.
4- It seems very weird to measure the MTT assay at 490 nm instead of 570 nm in that advanced device.
5- the method used for determination of the IC50 should be clearly stated because graphically the values seems away from the reported values in table 1
6- The raw data for determination of the platinum cytotoxicity are missed
7- The structure of triptolide should be drawn

Reviewer 3 ·

Basic reporting

The manuscript contains some spelling and grammatical a so it is better to be wholly revised by an English language expert.

Experimental design

no comments

Validity of the findings

no comments

Additional comments

Title: Discovery of potential targets of triptolide through inverse docking in ovarian cancer cells


The manuscript is somewhat interesting but certain points should be considered and are written below.
1. Regarding the introduction section, it should be more comprehensive with more details about the disease, its causes, methods of propylaxis and what previously used to alleviate the disease and what will this drug offer better than what previously reported.
2. The manuscript contains some spelling and grammatical a so it is better to be wholly revised by an English language expert.
3. The structure of the compound should be drawn and structural activity relationship should be comprehensively discussed
4. The references should be carefully checked to be all in the same style comprising the author’s name as some lack end pages.
5. Most of the figures are unclear

---

## Round 0.2 · Minor Revisions

Two specialists in the field re-evaluated your submission. One of the reviewers requested further improvements in English usage. I agree with him.

Reviewer 1 ·

Basic reporting

no comment

Experimental design

no comment

Validity of the findings

no comment

Additional comments

All the issues raised by this reviewer were successfully addressed. Consequently, the manuscript has been significantly improved and can be in its current version recommended for publication in PeerJ.

Reviewer 2 ·

Basic reporting

The English used is still not acceptable

Experimental design

The experimental design has much improved and corrected

Validity of the findings

Interesting findings

---

## Round 0.3 · accepted · Accept

In my view, the authors carried out all the modifications indicated by the reviewers. The paper can be accepted as it is.